# EMBO
### reports

# *scientific report*

# Introns of plant pri-miRNAs enhance miRNA biogenesis

*Dawid Bielewicz[1], Malgorzata Kalak[1], Maria Kalyna[2], David Windels[3], Andrea Barta[2], Franck Vazquez[3+], Zofia Szweykowska-Kulinska[1++] & Artur Jarmolowski[1+++]*

[1]Department of Gene Expression, Faculty of Biology, Adam Mickiewicz University, Poznan, Poland, [2]Max F. Perutz Laboratories, Medical University of Vienna, Vienna, Austria, and [3]Botanical Institute of the University of Basel, Zürich-Basel Plant Science Center, Part of the Swiss Plant Science Web, Basel, Switzerland

Since advance online publication the positioning of the labels in Figures 1 and 3 has been corrected.

Plant *MIR* genes are independent transcription units that encode long primary miRNA precursors, which usually contain introns. For two miRNA genes, *MIR163* and *MIR161*, we show that introns are crucial for the accumulation of proper levels of mature miRNA. Removal of the intron in both cases led to a drop-off in the level of mature miRNAs. We demonstrate that the stimulating effects of the intron mostly reside in the 5′ss rather than on a genuine splicing event. Our findings are biologically significant as the presence of functional splice sites in the *MIR163* gene appears mandatory for pathogen-triggered accumulation of miR163 and proper regulation of at least one of its targets.
Keywords: plant; Arabidopsis; RNA; splicing; miRNA

## INTRODUCTION

MicroRNAs (miRNAs) are 20–22-nt-long small RNAs that regulate the expression of genes involved in critical developmental programmes or in response to specific environmental conditions [1–5]. In plants, a set of 15–20 miRNA families that are evolutionarily highly conserved and serve in the regulation of crucial developmental programmes have been identified. In contrast, the other miRNA families are lineage- or species-specific, and serve in the regulation of specialized aspects of plant life [6]. While animal miRNAs are generally embedded into introns of protein-coding genes, plant miRNAs are encoded by independent *MIR* genes that are transcribed by Pol II to yield long primary miRNA precursors (pri-miRNAs) [7–9]. Processing of the pri-miRNAs occurs in two steps by the RNAse III enzyme DICER-LIKE1 (DCL1) and its main double-strand RNA-binding partner DRB1/HYL1 [10–12]. The first cut generates the intermediate hairpin-containing pre-miRNAs, whereas the second cut releases the miRNA/miRNA* duplexes. The miRNA strand is then incorporated into ARGONAUTE (AGO) effector complexes to guide RNA cleavage or translation inhibition [13–15]. The biogenesis and function of miRNAs are tightly regulated at several levels to ensure that proper regulation of the mRNA targets is maintained and adjusted in changing environmental conditions that includes posttranscriptional feedback regulation of *DCL1* and *AGO1* mRNA levels [16].

We have recently shown that plant pri-miRNAs are unexpectedly long and contain one or more introns located usually in their 3′ regions downstream of the hairpin folds that encode the miRNA duplexes [17,18]. Our laboratory and others have shown that these pri-miRNAs are bound, similarly to pre-mRNAs, by the nuclear cap-binding complex (CBC) proteins, CBP80 and CBP20, to promote their splicing in a step that involves the $C_2H_2$ zinc finger protein SERRATE, which is proposed to bridge the CBC and the spliceosome [17,19]. Our work suggests that splicing and processing of pri-miRNAs are coupled processes that might influence each other, although it is still unclear whether the two processes occur simultaneously or sequentially. Moreover, while the presence of introns appears to be a widespread conserved feature of plant *MIR* genes [17,18], their biological significance and the effect of pri-miRNA splicing on the biogenesis and function of miRNAs have not been evaluated.

We have introduced intron-less and splice site mutated versions of the single-copy gene *AtMIR163* in a miR163-defective mutant background, and similar *AtMIR161* versions were tested in transient expression assays in *Nicotiana benthamiana* to evaluate the relationship between splicing and processing of miRNA. Our data demonstrate that the introns of *MIR* genes are essential for the accumulation of proper levels of mature miRNAs. In addition, in the case of miR163, we also show that the intron is required for

[1]Department of Gene Expression, Faculty of Biology, Adam Mickiewicz University, Umultowska Street 89, 61-614 Poznan, Poland
[2]Max F. Perutz Laboratories, Medical University of Vienna, A-1030 Vienna, Austria
[3]Botanical Institute of the University of Basel, Zürich-Basel Plant Science Center, Part of the Swiss Plant Science Web, CH-4056 Basel, Switzerland
[+]Corresponding author. Tel: +41 61 2673517; Fax: +41 61 2672330;
E-mail: franck.vazquez@unibas.ch
[++]Corresponding author. Tel: +48 61 8295766; Fax: +48 61 8295949;
E-mail: zofszwey@amu.edu.pl
[+++]Corresponding author. Tel: +48 61 8295959; Fax: +48 61 8295949;
E-mail: artjarmo@amu.edu.pl

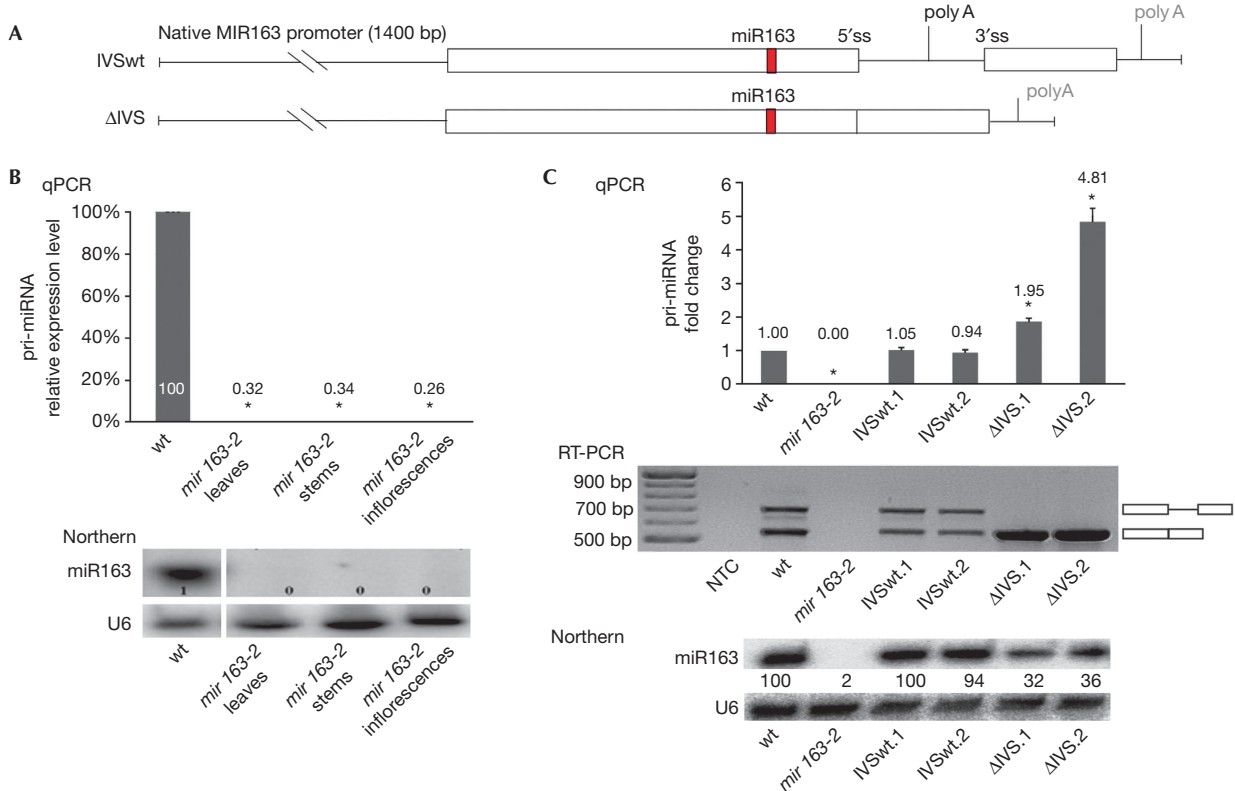

**Fig 1 |** The intron of *MIR163* stimulates the biogenesis of miR163. (**A**) Schematic representation of *MIR163* gene variants used. The splicing sites and position of proximal and distal poly(A) sites are shown. (**B**) Level of pri-miR163 (upper panel) and miR163 (lower panel) recorded in different organs of the *mir163-2* mutant. U6 snRNA serves as a loading control. (**C**) Level of pri-miR163 (upper panel), pri-miR163 splicing variants (middle panel) and miR163 (lower panel) recorded in lines expressing *MIR163* variants shown in **B**. IVSwt.1 and IVSwt.2 are independent lines with intron-containing *MIR163*; IVS.1 and ΔIVS.2 are independent lines with intron-less *MIR163*. U6 snRNA serves as a loading control. Error bars indicate s.d. (*n* = 3), and asterisks indicate a significant difference between the indicated sample and control wild type plants (Mann–Whitney test, *P* < 0.05). NTC, non-template control.

proper regulation of its newly validated mRNA target in response to the bacterial pathogen *Pseudomonas syringae* DC3000. Further genetic analyses showed that most of the stimulating effects of the intron reside in the 5′ splice site that might be caused by the binding of U1 snRNP that creates a connection between the spliceosome and the miRNA machinery. Analyses of several intron-containing *MIR*s in SR protein mutants allowed us to ascertain our conclusion concerning the direct requirement of introns for proper miRNA biogenesis.

## RESULTS AND DISCUSSION
### Intron stimulates the biogenesis of miR163
To evaluate the significance of introns in pri-miRNAs and specifically whether they influence the level of mature miRNAs, we introduced original or mutated *MIR163* constructs controlled by the native *MIR163* promoter in the *mir163-2* mutant (Fig 1A). This mutant contains a T-DNA insertion in the promoter of the intron-containing single-copy *MIR163* gene, and has undetectable levels of pri-miRNA163 in any of the tissues tested (Fig 1B, upper panel), as well as undetectable levels of mature miR163 (Fig 1B, lower panel). When the IVSwt construct, that is, a wild type (wt) *MIR163* gene copy, was introduced in the *mir163-2* mutant, the

level of miR163 was restored to the wt level in the two independent transgenic lines obtained (100% in IVSwt.1 and 94% in IVSwt.2; Fig 1C, lower panel). In contrast, the level of miR163 in the two ΔIVS lines expressing an intron-less *MIR163* gene was three times lower than that of wt plants (32% in ΔIVS.1 and 36% in ΔIVS.2; Fig 1C, lower panel). Interestingly, the level of pri-miR163 generated in ΔIVS.1 and ΔIVS.2 plants was significantly higher than that recorded in wt plants or in IVSwt plants (Fig 1C, upper panel). Thus, together these observations show that the intron of *MIR163* is required for accumulation of proper levels of miR163, and suggests that the processing of pri-miR163 is significantly altered by the absence of the intron in ΔIVS plants. We also obtained similar results for another miRNA, miR161, that suggests our conclusion that introns stimulate the biogenesis of plant miRNAs is also true for other *MIR* genes (supplementary Fig S2 online).

### Splice sites are required for proper miR163 biogenesis
The intron of *MIR163* is required for the accumulation of proper levels of miR163 but it is unclear whether this is caused by its splicing or by an unknown stimulatory feature of a sequence motif of the intron. To answer this, we generated *MIR163* variants in which the splice sites were mutated (Fig 2A). The two

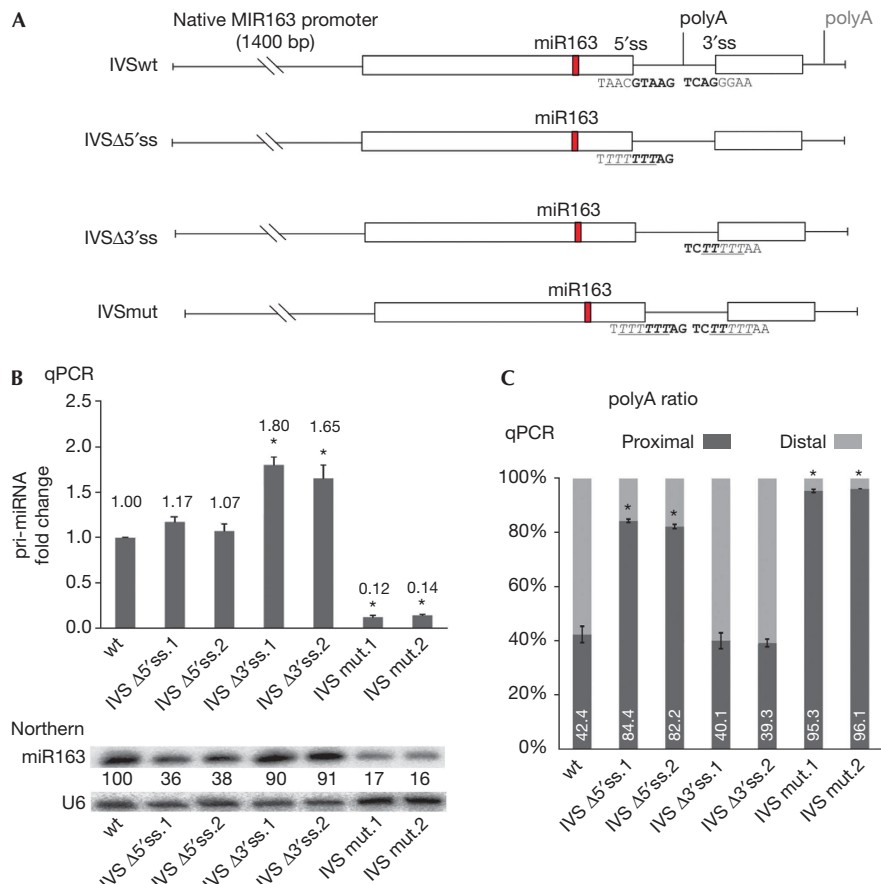

**Fig 2 | Mutations of splice sites in *MIR163* affects the biogenesis of miR163. (A)** Schematic representation of *MIR163* gene variants used. The splicing sites and position of proximal and distal poly(A) sites are shown. The sequences of wt and mutated splice sites are also shown. (**B**) Level of pri-miR163 splice variants (upper panel) and miR163 (lower panel) recorded in lines expressing *MIR163* gene variants. U6 snRNA serves as a loading control. (**C**) Ratio of proximal versus distal poly(A) site usage determined in wt plants and *MIR163* variant lines. IVSΔ5′ss.1 and IVSΔ5′ss.2 are independent lines in which the *MIR163* 5′ss was mutated; IVSΔ3′ss.1 and IVSΔ3′ss.2 are independent lines in which the *MIR163* 3′ss was mutated; IVSmut.1 and IVSmut.2 are independent in which the *MIR163* 5′ss and 3′ss are mutated. Error bars indicate s.d. ($n=3$), and asterisks indicate a significant difference between the indicated sample and control wt plants (Mann–Whitney test, $P < 0.05$). wt, wild type.

independent IVSmut transgenic lines, in which *MIR163* had both 5′ and 3′ splice sites mutated, displayed a strong decrease in miR163 accumulation (16–17% of wt level) associated with a strong decrease in pri-miR163 levels (12–14% of wt level; Fig 2B). This suggests that the IVSmut primary transcript is far less stable than the original pri-miRNA with functional splice sites.

In the IVSΔ5′ss lines, in which *MIR163* had only the 5′ splice site mutated, miR163 accumulated to only 36–38% of the wt level but the pri-miR163 level was similar to that of wt plants (Fig 2B). Importantly, in the IVSΔ3′ss lines, in which *MIR163* had only the 3′ splice site mutated, miR163 accumulated to levels similar to that of wt plants (90–91% of wt level; Fig 2B, lower panel). Moreover, the pri-miR163 accumulated 1.65–1.8-fold more in IVSΔ3′ss than in wt plants (Fig 2B, upper panel). These results show that the accumulation of miR163 is far less affected by mutation in the 3′ splice site than in the 5′ splice site. Moreover, mutation of both splice sites had an additive effect on decreasing the accumulation of miR163. Overall our data show that splicing, or at least the presence of the 5′ splice site, is important for

accumulation of proper levels of miR163. This conclusion was further strengthened by the similar results that we obtained with the *MIR161* construct containing a mutated 5′ss (supplementary Fig S2 online).

## Two poly(A) sites are used in the *MIR163* gene

Our 3′ RACE experiments showed that an alternative proximal poly(A) site located within the intron gives rise to a shorter *MIR163* transcript (Fig 2A). In the wt plants, this proximal poly(A) site is used for about 40% of the *MIR163* transcripts (Fig 2C). The frequency of alternative poly(A) transcripts in IVSwt plants was similar to that of wt non-transformed plants, whereas in intron-less ΔIVS plants only the distal poly(A) site was used as the proximal poly(A) site was removed together with the intronic sequence. We ascertained that the changes observed in IVSmut, IVSΔ5′ss and IVSΔ3′ss are due to defects in splicing by showing that spliced pri-miR163 was not detectable in these lines (supplementary Fig S1 online). We also determined that the ratio of proximal and distal poly(A)-tailed transcripts in these plants was altered in IVSΔ5′ss

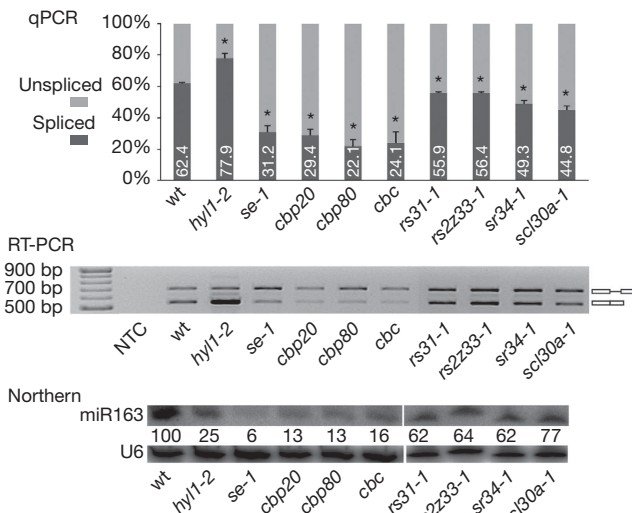

Fig 3 | SR proteins affect biogenesis of miR163. Efficiency of pri-miR163 splicing recorded by semiquantitative RT-PCR (upper panel) and by final-point RT-PCR (middle panel) in *sr* mutants (*rs31-1*, *rs2z33-1*, *sr34-1* and *scl30a-1*) and miRNA biogenesis mutants (*hyl1-2*, *se-1*, *cbp20*, *cbp80* and *cbc*, which is the *cbp20/cbp80* double mutant). The level of miR163 is given in the lower panel. U6 snRNA serves as a loading control. Error bars indicate s.d. (*n* = 3), and asterisks indicate a significant difference between the indicated sample and control wt plants (Mann–Whitney test, *P* < 0.05). NTC, non-template control.

and IVSmut lines compared with wt plants but not in IVSΔ3′ss (Fig 2C). The proximal poly(A) site was more frequently used in IVSΔ5′ss lines compared with the wt plants (40% in wt; more than 80% in IVSΔ5′ss). In IVSmut plants, in which *MIR163* had both 5′ and 3′ splice sites mutated, usage of the proximal poly(A) site was almost exclusive (more than 95%). Dreyfuss and coworkers have recently found that U1 snRNP can shield premature polyadenylation sites in human cells [20]. Therefore, it is possible that binding of U1 snRNP to the 5′ss of pri-miR163 might inhibit usage of the proximal poly(A) site. Importantly, our data also suggest a slight contribution of the 3′ss to this process (80% in IVSΔ5′ss and 95% in IVSmut). Earlier studies by Valcarcel and coworkers have shown that U2AF65, recognizing the 3′ss and/or U-rich sequences within introns, is able to stimulate interactions between U1 snRNP and the 5′ss [21]. Thus, similar to human cells, the 3′ss additive effects that we have observed in our experiments might be owing to the stimulation of U1 binding to the 5′ss similarly to that reported in human cells.

## SR proteins affect miR163 biogenesis
SR proteins are important splicing factors that act as positive regulators of splicing. To test whether excision is important in the intron effect that we have observed on miR163 biogenesis, we used different *sr*-null mutants. The level of miR163 was decreased to 62–77% of the wt level in many of the SR protein mutants tested (Fig 3, lower panel). Moreover, these differences correlated with changes in the splicing efficiency of pri-miR163 (Fig 3, upper and middle panels). These effects were similar to those observed in the splicing-miRNA biogenesis mutants *se-1*, *cbp20*, *cbp80* and in the *cbp20/cbp80* double mutant, *cbc*, in which the level of miR163

is dramatically reduced (Fig 3, lower panel) and the splicing efficiency is affected (Fig 3, upper and middle panels) [19].

Importantly, the effect of SR proteins on miRNA levels was also observed for other intron-containing *MIR* genes (supplementary Fig S2 online), but not all SR protein mutants tested showed changes in pri-miRNA splicing and miRNA accumulation. Although we provide evidence for the involvement of plant SR proteins in the biogenesis of miRNA from intron-containing genes, changing SR protein expression levels had much weaker effects on the accumulation of miR163 than the mutations in the IVSmut (5′ss and 3′ss mutated) or IVSΔ5′ss constructs. Nevertheless, these observations strongly support our previous conclusion that splicing stimulates miRNA production from intron-containing pri-miRNAs, and that the functional connection between intron removal and miRNA accumulation relies on the recognition of the 5′ss rather than on genuine intron excision.

## miR163 targets a SAM-dependent methyltransferase
To experimentally validate some of our *in silico* predicted miR163 mRNA targets, we carried out 5′RLM-RACE experiments in wt and *mir163-2* mutant plants, as well as in the *xrn4-3* mutant that accumulates higher levels of 3′ cleavage fragments of selected miRNA targets [22]. The 5′ RACE amplification product for At1g66690, which encodes an *S*-adenosyl-L-methionone-dependent methyltransferase, was at the expected size, and accumulated to slightly higher levels in the *xrn4-3* mutant than in the wt plant (Fig 4A, left panel). Moreover, in the *mir163-2* mutant, this product was not observed, and instead a ladder of different RNA degradation fragments was detected (Fig 4A, right panel). The cloning and sequencing of this 5′ RACE product identified the cleavage site guided by miR163 (Fig 4A, upper sequence). Furthermore, the steady-state level of At1g66690 mRNAs in the miRNA biogenesis mutants *hyl1-2*, *se-1*, *cbp20*, *cbp80* and *cbc*, the double mutant *cbp20/cbp80*, was higher than in wt plants (Fig 4B). This increased mRNA accumulation was in agreement with the decreased miR163 level in the miRNA biogenesis mutants tested (Fig 3, lower panel). Thus, taken together our data show that At1g66690 mRNA is a target of miR163. Two more targets of miR163 have been already described, At1g66700 and At3g44860, of which the former also belongs to the family of *S*-adenosyl-L-methionine-dependent methyltransferases [23].

## Induction of miR163 depends on functional splice sites
Recent work has shown that miR163 accumulation is induced by various biotic stress [23]. To test whether the stimulatory effect of an intron on miR163 biogenesis is biologically significant, we compared the response of wt and IVSmut plants to infection by *P. syringae*. Interestingly, the level of miR163 in wt plants reached 135% and 145% of the wt level after 24 and 72 h of infection (Fig 4C, middle panel), whereas the level of the At1g66690 mRNA target correlated by a simultaneous reduction (Fig 4C, lower panel). In contrast, in the IVSmut plants miR163 accumulation was low and uninduced after 24 or 72 h of infection (Fig 4C, middle panel), although the level of pri-miR163 was increased at these time points (Fig 4C, upper panel). These observations show that the posttranscriptional regulation of miR163 biogenesis under biotic stress condition is impaired in IVSmut plants, and highlight an important role for the *MIR163* intron and its

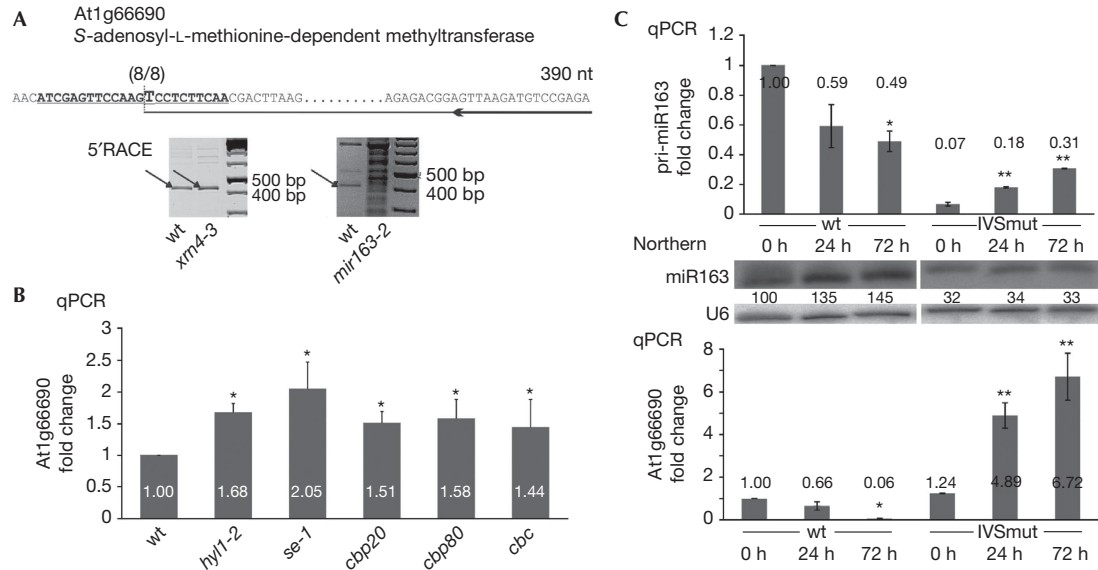

**Fig 4** | The accumulation of miR163 after pathogen infection depends on the presence of functional splice sites in the *MIR163* gene. (**A**) Identification of miR163-guided cleavage sites in At1g66690 mRNAs. The cleavage site was detected in wt and in the *xrn4-3* mutant. The PCR products at the expected size are marked with arrows. The upper sequence shows the exact cleavage site identified and the sequence of the primer used for 5′RACE. The number of clones (8/8) sequenced are given. (**B**) Level of the At1g66690 mRNA recorded in wt and in the miRNA biogenesis mutants: *hyl1-2*, *se-1*, *cbp20*, *cbp80 and cbc* (that is, *cbp20/cbp80* double mutant). Error bars indicate s.d. (*n* = 3), and asterisks indicate a significant difference between the indicated sample and control wt plants (Mann–Whitney test, *P* < 0.05). (**C**) Level of pri-miR163 (upper panel), of miR163 (middle panel), and of its validated mRNA target (lower panel) recorded in wt and IVSmut plants after 0, 24 or 72 h infection with *P. syringae* DC3000. Error bars indicate s.d. (*n* = 3). One asterisk indicates a significant difference between the indicated sample and wt plants at the 0 h time point, and double asterisk indicates a significant difference between the indicated sample and IVSmut plants at the 0 h time point (Mann–Whitney test, *P* < 0.05). wt, wild type.

functional splice sites in regulation of miR163 biogenesis during bacterial infection. The microarray data have already suggested that the expression of At1g66690 is connected with bacterial infection of Arabidopsis [24]. Owing to the type of microarrays used in the experiments (Affymetrix ATH1), it was unclear if the effect observed (the authors claimed that the level of the At1g66690 transcript increased after infection) meant the level of full-length transcripts and/or stable 3′ fragments of miRNA-directed cleaved mRNA. Since we as well as others observed the increased accumulation of miR163 on *P. syringae* infection [25], and miR163 is involved in cleavage of At1g66690 transcripts, the accumulation of target 3′ fragments can explain the microarray results [24].

## CONCLUSIONS

Our experiments have shown that the intron of *MIR163* is required for proper biogenesis and function of its mature miRNA. The removal of the *MIR163* intron, as well as mutations that block its splicing, led to a significant reduction in the level of mature miR163 and to accumulation of its newly validated target. The disruption of the 5′ss had a stronger impact on miR163 accumulation than the disruption of the 3′ss had. However, the disruption of the 3′ss in addition to the 5′ss led to an even higher decrease in the level of mature miR163. Although we could not exclude a direct effect of splicing on stimulation of miR163 biogenesis, our data indicate that the crosstalk between the spliceosome and the miRNA biogenesis machinery most likely involves recognition of the 5′ss by the U1 snRNP. Importantly, our

similar analyses of *MIR161* in transient expression assays in *N. benthamiana*, as well as analyses of several intron-containing pri-miRNAs in SR protein mutants strengthen and expand our conclusion that pri-miRNA introns have a direct effect on proper biogenesis of miRNAs. Interestingly, a feed-forward model of the crosstalk between miRNA biogenesis and splicing in mammals has been recently proposed [26]. In contrast to plants, most of the miRNAs in mammals are encoded by introns of protein-coding genes. It has been shown that in this type of miRNA genes, U1 snRNP recognizes the 5′ss and promotes recruitment of Drosha, a human RNase III-type enzyme that catalyses the excision of miRNA precursors from pre-mRNAs. According to the model proposed, the U1 snRNP first recognizes the 5′ss of the miRNA-containing intron, which leads to increased efficiency of the enzymatic activity of Drosha. On the other hand, Drosha bound to the cut intron generates a better splicing substrate by stabilizing U1 snRNP binding to the 5′ss, thus subsequently promoting splicing completion. Although the mechanism of functional connections between splicing and miRNA biogenesis in plants has to be different, as most plant miRNAs are generated from long non-coding precursors, our results strongly suggest the involvement of U1 snRNP in such crosstalk that is similar to the feed-forward model proposed for mammalian miRNAs embedded in introns. Schwab and colleagues report in this issue of EMBO *reports* a similar positive effect of introns on miRNA accumulation that corroborates our conclusions. However, a difference lies on the effect of 5′ss mutations, which in their study either lead to no change in miR163 accumulation or

to increased accumulation of miR172. This discrepancy might be due to mutations introduced into the mutated 5'ss, which do not sufficiently compromise (miR163) or even favour (miR172) the binding of U1 snRNP. It is also possible that the difference arised from the use of promoters with different strength: the native *MIR*163 promoter in our experiments, and the CaMV 35S promoter in Schwab and colleagues' studies. This hypothesis is actually supported by our observations with the 5'ss mutated version of the *MIR161* gene, which is driven by the 35S promoter, leads to weaker effects on miRNA biogenesis than the 5'ss mutated version of *MIR163*, which is driven by its native promoter (compare the results presented in Fig 2 and supplementary Fig S2 online). Future work will have to test these possibilities in detail and provide further understanding of this specific regulatory mechanism.

## METHODS

**Plant material.** Arabidopsis seeds were stratified on ½ MS or directly on soil for 2 days at 4 °C and grown at 22 °C with 16-h light in SANYO MLR-351H growth chamber. The *mir163-2* mutant (SALK_0034556) was identified by PCR (oligonucleotides are listed in supplementary Table S1 online). The SR protein knock-out lines used in this study are: SALK_106067 (*sr34-1*), SALK_021332 (*rs31-1*), GABI_180D12 (*rs2z33-1*) and SALK_095431 (*scl30a-1*).

**Generation of transgenic lines.** *MIR163* gene variants were prepared by PCR (oligonucleotides are listed in supplementary Table S1 online), and cloned in pENTR/D-TOPO plasmid (Life Technologies) using *Not*I and *Asc*I restriction sites. The sequence of all constructs used were verified by sequencing. To perform the Gateway LR reaction, the pENTR/D-TOPO plasmids containing inserts were digested with *Pvu*II (Fermentas). For expression in plants, pMDC99 or pMDC123 Gateway binary vectors were used [27]. Transgenes were introduced in *mir163-2* plants by the *Agrobacterium*-mediated floral dip transformation [28].

**Transient expression in *N. benthamiana*.** *MIR161* gene variants were prepared by PCR (oligonucleotides are listed in supplementary Table S1 online), and cloned in pCR8 plasmid (Life Technologies). For expression in plants, pMDC32 Gateway binary vector was used [27]. For one construct, four leaves of 5-week-old *N. benthamiana* were infiltrated with *Agrobacterium tumefaciens* ($OD_{600} = 0.6$ in 10 mM MES pH 5.6, 10 mM $MgCl_2$). Leaves were harvested after 72 h.

**RNA isolation and analysis.** Total RNA from 10-day-old seedlings or from leaves of 21-day-old plants was isolated using Trizol reagent (Life Technologies) and treated with Turbo DNase (Ambion) before reverse transcription with oligo $dT_{(18)}$ (Fermentas) and SuperScript III Reverse Transcriptase (Life Technologies). We analysed sRNA as described previously [17]. To perform 5'RLM-RACE, the GeneRacer kit (Life Technologies) was used according to the manufacturer's instructions.

**Quantitative real-time PCR analysis (qPCR).** Real-time qRT-PCRs were performed as described previously [29]. Estimation of poly(A) site usage was done with two real-time PCR reactions each detecting one of the pri-miRNA163 isoform. The two reactions had equivalent efficiencies that allowed to calculate the relative abundance of each isoform (supplementary Fig S3 online).

**Bacteria treatment.** Two weeks old plants were sprayed with *P. syringae* DC3000 at an $OD_{600} = 0.2$ in 10 mM $MgCl_2$. *P. syringae* was grown overnight in YEB medium at 28 °C. Aerial parts of the plants were collected just after treatment or after 24 and 72 h.

**Statistics.** Statistical tests were performed using MS-Excel 2007 and the Statistica program. The Mann–Whitney U-test was used. *P*-values are presented in figure legends.

ACKNOWLEDGEMENTS
This work was supported by the Polish Ministry of Science and Higher Education (3011/B/P01/2009/37), the Polish National Science Center NCN (UMO-2011/01/M/NZ2/01435 to A.J., and UMO-2012/04/M/NZ2/00127 to Z.S.-K.), and by The Rectors' Conference of the Swiss Universities (Sciex Project 11.115 to D.B., F.V. and Z.S.-K.). A.B. was supported by the Austrian Science Fund FWF (SFB 1710, 1711). F.V. was supported by the Swiss National Science Foundation (PZP00P3_126329 and PZP00P3_142106). The PhD fellowship of D.B. is part of the International PhD Programme 'From genome to phenotype: A multidisciplinary approach to functional genomics' (MPD/2010/3) funded by the Foundation for Polish Science (FNP). We thank Julian I. Schroeder and Csaba Koncz for sharing seeds of the *cbp80* and *cbp20* mutants, respectively. We thank Rebecca Schwab, Sascha Laubinger and Olivier Voinnet for scientific discussions and friendly cooperation. We also thank Paul Zalesky for critical reading of the manuscript, and correcting language errors.

*Author contributions*: D.B. designed and carried out the experiments, analysed and discussed the results, and helped in manuscript preparation; Malgorzata K. characterized the *mir163-2* mutant; Maria K. and A.B. generated SR protein mutants and discussed the results; D.W. helped with experiments involving *N. benthamiana* transient expression assays; F.V., Z.S.-K. and A.J. designed the experiments, discussed the results and wrote the paper.

CONFLICT OF INTEREST
The authors declare that they have no conflict of interest.

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
