## [Review Process File · EMBO Reports]

Manuscript EMBOR-2012-36796

Introns of plant pri-miRNAs enhance miRNA biogenesis and function.

Dawid Bielewicz, Malgorzata Kalak, Maria Kalyna, David Windels, Andrea Barta, Franck Vazquez, Zofia Szweykowska-Kulinska and Artur Jarmolowski

*Corresponding authors: Artur Jarmolowski, Adam Mickiewicz University
Zofia Szweykowska-Kulinska, Adam Mickiewicz University
Franck Vazquez, Botanical Institute University of Basel*

Review timeline:	Submission date:	31 October 2012
	Editorial Decision:	26 November 2012
	Revision received:	26 February 2013
	Editorial Decision:	15 March 2013
	Revision received:	09 April 2013
	Accepted:	12 April 2013

Editor: Esther Schnapp

Transaction Report:

1st Editorial Decision

26 November 2012

Thank you for the submission of your manuscript to EMBO reports. We have now received the full set of referee reports on your study that is copied below.

As you will see, all referees acknowledge that the findings are potentially interesting and suitable for publication in EMBO reports. However, both referees 1 and 2 point out that the observed effect of introns on enhancing mature miR163 levels should be extended to other miRNAs. Both referees also mention the inconsistency between your study and the one from Voinnet's lab regarding the requirement of the 5' splice site of introns for the enhanced levels of mature miRNAs. This inconsistency must be addressed and at least a possible explanation should be provided. Referee 2 suggests that the constructs used in both studies should be compared in order to address the opposing results. This referee finally also indicates missing statistical analyses of the data that need to be performed.

Given these referee comments, we would like to invite you to revise your manuscript with the understanding that the referee concerns must be fully addressed and their suggestions (as detailed above and in their reports) taken on board. Acceptance of the manuscript will depend on a positive

outcome of a second round of review and I should also remind you that it is EMBO reports policy to allow a single round of revision only and that, therefore, acceptance or rejection of the manuscript will depend on the completeness of your responses included in the next, final version of the manuscript.

Revised manuscripts should be submitted within three months of a request for revision; they will otherwise be treated as new submissions. Also, the revised manuscript may not exceed 30,000 characters (including spaces and references) and 5 figures plus 5 supplementary figures, which should directly relate to the corresponding main figure. Please also include scale bars in all microscopy images, the number (n) of experiments and please specify the error bars and statistical tests used to calculate p-values for all quantifications in the corresponding figure legends.

We also recently decided to offer the authors the possibility to submit "source data" with their revised manuscript that will be published in a separate supplemental file online along with the accepted manuscript. If you would like to use this opportunity, please submit the source data (for example entire gels or blots, data points of graphs, additional images, etc.) of your key experiments together with the revised manuscript.

As part of the EMBO publication's Transparent Editorial Process, EMBO reports publishes online a Review Process File to accompany accepted manuscripts. This File will be published in conjunction with your paper and will include the referee reports, your point-by-point response and all pertinent correspondence relating to the manuscript.

I look forward to seeing a revised version of your manuscript when it is ready.

REFeree REPORTS:

Referee #1:

In this manuscript, Bielwicz et al. report that the intron of MIR163 is essential for the biogenesis of miR163 and its function during biotic stress. By expressing the original (with intron) or mutated (intron-less) MIR163 constructs in MIR163 T-DNA knock-out lines, they showed that the processing of pri-miR163 to mature miR163 was significantly reduced in lines expressing the intron-less MIR163. The 5' splice site of the intron was more important than 3' splice sites for the intron effect on the proper accumulation of miR163. They also showed that mutation of splicing factor SR proteins led to decreased level of mature miR163, suggesting that the splicing and pri-miRNA processing were coupled processes which may influence each other. Furthermore, the intron of miR163 played an important role in induced miR163 accumulation during bacterial infection, as transgenic lines expressing intron-less MIR163 showed impaired miR163 biogenesis with reduced miR163 levels.

This study provided a new insight into the roles of introns in intron-containing plant MIRNA genes. The experiments were well-designed with proper controls and replication in multiple transgenic lines. The data were clearly presented in quantitative ways for easy comparison, and it was very clear to see the pri miRNA along with the mature miR163 level. While the main focus of the paper was on miR163 and the data were solid and convincing enough to be published in EMBO Reports, the results left me wondering whether the intron effect on miRNA biogenesis would hold true for other intron-containing miRNAs. In other words, the finding will have more profound impact if the authors had demonstrated the critical roles of intron in miRNA biogenesis for more than one

miRNA.

Below is a summary of comments/concerns about the manuscript.

1. Page 2, Introduction "The miRNA strand is then incorporated into ARGONAUTE (AGO) effector complexes to guide RNA cleavage, translation inhibition, or DNA methylation in some cases". The authors may want to clarify the sentence as majority of plant miRNA activity is through RNA cleavage and translational inhibition. The role of miRNAs on DNA methylation is mostly indirect, through siRNA production, as demonstrated in reference 16.
2. Page 4, 1st paragraph, the author claimed that "binding of U1 snRNP to the 5' splice site of pri-miR163 might inhibit usage of proximal poly(A) site". Since there is no data supporting this, there is not sufficient data to draw this conclusion.
3. Page 4, 2nd paragraph, the author referred to Figure 4, but did not indicate which panel (Panel A, B, or C?). Please specify.
4. Figure 3 and Figure 3 legend: in the legend, the authors listed four sr mutants. sr31-1 was listed before the overexpressing line (OX), which made the figure very confusing. I suggest that the author sub-divide the mutants in several categories to make the figure easier to follow.
5. Page 4, 3rd paragraph "...instead a ladder of different RNA degradation fragments was detected (Fig. 4B)", it should be Fig. 4A.
6. It was interesting that splicing and pri miRNA processing may influence each other. Besides mutants in splicing factors, have the authors tried to tease apart these two factors such as testing the constructs in mutant defective in miRNA processing?
7. Have the authors looked into the attributes of introns (sequences, length etc) which could possibly contribute to their ability to regulate miRNA processing?
8. I found the figures to be a bit hard to read in their formatting - the font sizes are quite small in some cases, particularly in the scale at which they'll be printed. The authors should improve the clarity of the presentation and artistry of the figures as it would greatly help the reader in understanding the data that they're showing.

Referee #2:

In the manuscript entitled "Introns of plant pri-miRNAs are required for proper biogenesis and function of miRNAs" the authors show that the introns of the MIR163 precursor is crucial for the accumulation of mature miR163 and that the splice sites are required for proper generation of miR163. Furthermore, they present data showing that several SR protein splicing factors affect generation of miR163. Finally, they show that miR163 is induced upon infection with *Pseudomonas syringae*. Overall, the manuscript addresses an interesting and timely topic and is well written.

To investigate the effect of the intron in the MIR163 gene on miR163 accumulation, the authors made use of a MIR163 T-DNA line. A construct containing the MIR163 gene with intron complements this mir163-2 mutant, whereas an intron-less MIR163 gene does not. In the legend to the corresponding Fig 1 the sentence about the U6 loading control refers to panel B, not panel A.

When the authors mutated the 5' splice site and 3' splice site of the MIR163 intron, they observed lower miR163 accumulation.

They also observed lower levels of priMIR163. They conclude that the mutant primary transcript is less stable than the wt precursor with functional splice sites.

When the intron-less priMIR163 was introduced into the mutant, the level of the priMIR163 was higher (Fig 1C). This could be due to reduced processing. In the light of the conclusion that the primary transcript with 5' and 3' splice sites mutated is less stable the question arises whether the stability of the intron-less priMIR163 could also be altered?

Furthermore, using constructs with either the 5' splice site or 3' splice site mutated individually the authors conclude that the 5' splice site has a larger impact on miR163 accumulation.

These results are congruent with results of the accompanying paper by Schwab and co-workers, as these authors also find that the stimulatory effect of introns on miRNA biogenesis resides in specific sequences within the intron rather than the splicing event. However, there is a major discrepancy to the findings in the accompanying paper:

While in the present manuscript the authors found an effect of mutating the 5' splice site, in Schwab et al no change in miR163 levels or miR172a levels was observed. This should be addressed perhaps by reciprocally comparing the constructs used in both studies.

Furthermore, the conclusion of Bielewicz and coworkers should be strengthened by analyzing other miRNA precursors to see whether the same effect could be observed for other priMIRs as well.

To support their conclusion that introns are required for miRNA biogenesis the authors investigated the miR163 level in mutants impaired in several SR proteins.

It seems that over-expressors of RS2Z33 and RS2Z33 mutants have the same effect on the miR163 level. The authors should comment on this and provide data on the protein levels compared to wt.

Furthermore, the effect in the sr mutants appears to be a higher accumulation of both the priMIR and the miR163 compared to wt plants rather than changes in the ratios of AS forms (Fig 3)?

In the absence of information on statistical significance, Fig 3 does not seem to support the conclusion that SR factor dependent splicing affects miR163 levels.

In the corresponding section of the manuscript, Fig 4 lower panel should read Fig 3 lower panel

Finally, the authors show that miR163 is induced upon infection with *Pseudomonas syringae* and the corresponding target, S-AdoMet dependent methyltransferase is concomitantly reduced.

The authors should discuss whether this has been found in previous large scale approaches to screen for miRNAs with altered levels upon treatment with bacterial pathogens.

Overall assessment according to EMBO REPORT scoring sheet:

- 1) The manuscript reports two findings related to the connection of miRNA processing and splicing and one finding related to the miRNA studied in this report: no
- 2) yes
- 3) The work points to a connection of miRNA processing and splicing in Arabidopsis which has not been explored in much detail so far: yes
- 4) Including the controls as detailed above does not necessarily lead to a longer manuscript.

Referee #3:

In this manuscript, Bielewicz et al find that the intron of MIR163 is essential for the accumulation mi163. They show that mutations abolishing 5' splicing site but not 3' splicing site disabled the function of intron, indicating that the 5' splicing site rather than splicing events is required for miR163 accumulation. The authors also showed that SR proteins also involves in accumulation miR163, providing further support for their conclusion. The finding is novel and is of interests of the field.

1, A concern is that several rather than one nucleotides are mutated that may complicate the interpretation of results.

2, The SR protein results did not provide support for the conclusions, as its effect may be indirect by affecting processing factors. This section may be removed.

There is an inconsistency in these two manuscripts. Bielewicz et al show that mutated 5' splicing site affects miRNA accumulation, while schwab find that mutated 5' splicing site does not. If possible, the authors shall provide explanation for this inconsistency, which may provides mechanic insight on how 3' intron affects miRNA accumulation.

The answers to referees comments and suggestions

Referee #1

First we want to thank the referee for recognizing the importance of our work. His constructive comments were very useful to prepare our revised manuscript and to expand the scope of our conclusions.

Comment 1: Page 2, Introduction "The miRNA strand is then incorporated into ARGONAUTE (AGO) effector complexes to guide RNA cleavage, translation inhibition, or DNA methylation in some cases". The authors may want to clarify the sentence as majority of plant miRNA activity is through RNA cleavage and translational inhibition. The role of miRNAs on DNA methylation is mostly indirect, through siRNA production, as demonstrated in reference 16.

Answer 1: According to this referee's suggestion we removed from the Introduction section the sentence fragment on the involvement of miRNA in DNA methylation. We agree, the mechanism of participation of miRNA in DNA methylation is not so clear as we have mentioned on it in the text. To describe this issue properly, much longer explanation would be needed. Since there is a space limit for an EMBO reports manuscript, we omit the description of the involvement of miRNAs in DNA methylation. Therefore, we removed also the reference 16 from the manuscript reference list.

Comment 2: Page 4, 1st paragraph, the author claimed that "binding of U1 snRNP to the 5' ss of pri-miR163 might inhibit usage of proximal poly(A) site". Since there is no data supporting this, there is not sufficient data to draw this conclusion.

Answer 2: We changed this sentence according to the referee's suggestion. It was before: "It suggest that binding...", and now is: "Therefore, it is possible that binding...". We would like to keep this suggestion in such form since it is a part of our discussion, and not the final statement based on our results.

Comment 3: Page 4, 2nd paragraph, the author referred to Figure 4, but did not indicate which panel (Panel A, B, or C?). Please specify.

Answer 3: We corrected this mistake in the text.

Comment 4: Figure 3 and Figure 3 legend: in the legend, the authors listed four sr mutants. sr31-1 was listed before the overexpressing line (OX), which made the figure very confusing. I suggest that the author sub-divide the mutants in several categories to make the figure easier to follow.

Answer 4: We have removed the RS2Z33 overexpressor from this paper (for the reasons see our answer to the referee #2. Figure 3 contains now only mutant lines and thus does not require any special sub-division.

Comment 5: Page 4, 3rd paragraph "...instead a ladder of different RNA degradation fragments was detected (Fig. 4B)", it should be Fig. 4A.

Answer 5: We corrected this mistake in the text.

Comment 6: It was interesting that splicing and pri-miRNA processing may influence each other. Besides mutants in splicing factors, have the authors tried to tease apart these two factors such as testing the constructs in mutant defected in miRNA processing?

Answer 6: We are currently doing crosses of Arabidopsis plants carrying our *MIR163* gene variants with various miRNA biogenesis as well as with selected splicing Arabidopsis mutants. The analyses of expression of different *MIR163* variants in these various genetic background should throw light on the mechanism of the crosstalk studied. The results of these experiments will be described in the next paper. However, we have already data showing interactions between SERRATE (a plant protein crucial for miRNA biogenesis) with some U1 snRNP proteins, as well as SERRATE and CBP80 (a subunit of the nuclear cap-binding protein complex, CBC), which support our idea that the interaction between CBC, SERRATE and U1 plays a crucial role as a specific backbone in functional connections of splicing and miRNA biogenesis.

Comment 7: Have the authors looked into the attributes of introns (sequences, length etc) which could possibly contribute to their ability to regulate miRNA processing?

Answer 7: We have tried to search for such elements but up to now we have been unable to find any characteristic attributes of the pri-miRNA introns tested. We are continuing such *in silico* approaches using advanced tools and novel strategies.

Comment 8: I found the figures to be a bit hard to read in their formatting - the font sizes are quite small in some cases, particularly in the scale at which they'll be printed. The authors should improve the clarity of the presentation and artistry of the figures as it would greatly help the reader in understanding the data that they're showing.

Answer 8: We did our best to improve the quality of the figures and make them easier to understand.

Referee #2

We want to thank the referee for her/his constructive comments. They were very useful to prepare our revised manuscript.

Comment 1: In the legend to the corresponding Fig. 1 the sentence about the U6 loading control refers to panel B, not panel A.

Answer 1: We corrected the mistake according to the referee's comment.

Comment 2: When the intron-less priMIR163 was introduced into the mutant, the level of the priMIR163 was higher (Fig 1C). This could be due to reduced processing. In the light of the conclusion that the primary transcript with 5' and 3' splice sites mutated is less stable the question arises whether the stability of the intron-less priMIR163 could also be altered?

Answer 2: We think that the lack of both splice sites is a clear signal for the degradation machinery to remove rapidly any miRNA precursor that cannot be spliced. The intron-less pri-miRNA, on the other hand, does not contain any intronic sequences, thus is recognized as a precursor after splicing, which should be further processed and therefore protect from degradation.

Comment 3: While in the present manuscript the authors found an effect of mutating the 5' splice site, in Schwab et al no change in miR163 levels or miR172a levels was observed. This should be addressed perhaps by reciprocally comparing the constructs used in both studies.

Answer 3: We confirmed that the 5'ss is more important than the 3'ss for stimulatory effect on miRNA biogenesis using another miRNA, miR161, and another expression system, agroinfiltration of *Nicotiana benthamiana* leaves (see Fig. S2 online). We realized, however, that this effect in our transient expression assay is weaker than in the transgenic lines tested. We think that the discrepancy between our results and those described by Schwab and colleagues is due to the promoter used: the native *MIR163* promoter versus the very strong CaMV 35S promoter. However, to get the final answer further analyses are needed, and we are performing such experiments.

Comment 4: Furthermore, the conclusion of Bielewicz and coworkers should be strengthened by analyzing other miRNA precursors to see whether the same effect could be observed for other priMIRs as well.

Answer 4: In addition to miR163 we carried out similar experiments on miR161, and showed that also the intron of *MIR161* stimulated the miR161 biogenesis. Moreover, our new results, added to the revised version of the manuscript, confirmed that the 5'ss mutation in the *MIR161* gene has a stronger effect on miR161 production than the mutated 3'ss. These results were obtained using the *N. benthamiana* agroinfiltration system. Thus, our conclusion that introns stimulate the biogenesis of plant miRNAs is also true for other *MIR* genes.

Comment 5: It seems that over-expressors of RS2Z33 and RS2Z33 mutants have the same effect on the miR163 level. The authors should comment on this and provide data on the protein levels compared to wt.

Answer 5: The RS2Z33 overexpressor line was created by expressing the RS2Z33 genomic clone (containing all the introns of the gene) under the strong 35S CaMV promoter (*35S:gatRSZ33*, see Maria Kalyna, et al. Mol Biol Cell. 2003;14(9):3565-3577). We have shown that RS2Z33 regulates splicing of its own pre-mRNA, which results in the preferential accumulation of non-coding RS2Z33 transcripts. During our further studies using these transgenics we have noticed that in later generations they resembled the *rs2z33* mutant plants. Therefore, we checked the splicing pattern of RS2Z33 in the *35S:gatRSZ33* plants used in this study. As shown in the figure below, levels of the fully spliced transcript (FS) coding for the RS2Z33 protein is lower in these plants than in wild type. That is why we consider now these plants as a knock-down for RS2Z33, and not the RS2Z33 overexpressor.

Though we can explain why the RS2Z33 overexpressor and the *rs2z33* mutant plants might have similar effects on the miR163 levels, we decided to remove the RS2Z33 overexpressor from this paper to avoid any confusions.

Comment 6: Furthermore, the effect in the *sr* mutants appears to be a higher accumulation of both the priMIR and the miR163 compared to wt plants rather than changes in the ratios of AS forms (Fig 3)? In the absence of information on statistical significance, Fig. 3 does not seem to support the conclusion that SR factor dependent splicing affects miR163 levels.

Answer 6: We added to Fig. 3 standard deviation values based on three different experiments performed. The results are fully reproducible. Our complimentary data on the influence of SR proteins on pri-miRNAs without introns showed non or very weak effects of these splicing factors on miRNA biogenesis from intron-less genes. Thus, there is definitely a clear connection between the levels of several SR proteins, splicing and miRNA accumulation, although, we agree, the effect can be in some cases indirect. We think, however, that the results showing the miR163 underaccumulation in various *sr* mutants support our final conclusion.

Comment 7: In the corresponding section of the manuscript, Fig 4 lower panel should read Fig 3 lower panel.

Answer 7: We corrected this mistake.

Comment 8: The authors should discuss whether this has been found in previous large scale approaches to screen for miRNAs with altered levels upon treatment with bacterial pathogens.

Answer 8: Yes, the involvement of At1g66690 in the bacterial infection response in Arabidopsis can be observed in some microarray experiments (for example in Mohr PG, Cahill DM (2007) Suppression by ABA of salicylic acid and lignin accumulation and the expression of multiple genes, in Arabidopsis infected with *Pseudomonas syringae* pv. tomato. *Funct. Integr. Genomics* 7(3): 181-91). The problem with the data presented by Mohr and Cahill is that their analyses were based on ATH1 Affymetrix array data. In addition, these authors do not show any results on miRNA expression, so any correlations between miRNA levels and their targets can be proposed. Because of the type of microarrays used in the paper by Mohr and Cahill, it is unclear if the effect (they claimed that the level of the At1g66690 transcript increased) observed means the level of full length transcripts and/or stable 3' fragments of miRNA-directed cleaved mRNA. The increased accumulation of miR163 upon *Pseudomonas syringae* infection observed by Zhang and colleagues (Zhang et al. (2011) Bacteria-responsive microRNAs regulate plant innate immunity by modulating plant hormone networks. *Plant Mol. Biol.* 75(1-2):93-105) might support the later possibility. In our manuscript we used real time PCR to monitor the level of full length At1g66690 transcript. The high throughput data indicate that the expression of At1g66690 is directly connected with bacterial infection of Arabidopsis, and our data establish a clear correlation between the levels of miR163 and one of its targets, At1g66690.

Referee #3

We want to thank the referee for recognizing the novelty of our findings. The comments were very useful to us to prepare the revised version of our manuscript.

Comment 1: A concern is that several rather than one nucleotides are mutated that may complicate the interpretation of results.

Answer 1: We are planning to do more complex mutagenesis of the *MIR163* 5' splice site to study the influence of these mutations on the pri-miR163 splicing and miR163 biogenesis. We are also going to control in these experiments the binding of U1 to the mutated 5'ss by the RIP method.

Comment 2: The SR protein results did not provide support for the conclusions, as its effect may be indirect by affecting processing factors. This section may be removed.

Answer 2: The lower production of miRNAs from intron-containing precursors in several *sr* mutants was reproducible in many experiments. We agree with the referee we cannot rule out the possibility that some of the effects seen are indirect. We established that the effect of SR proteins on accumulation of miRNAs originated from genes without introns is very weak if any. Thus, the effects of SR proteins on miRNAs maturation described for miR161, miR163 and miR171 in this manuscript is connected with introns and/or splicing. Therefore, we would like to keep Fig. 4 in the manuscript, because the data support our observations that introns in pri-miRNAs are required for proper accumulation of mature miRNAs.

Comment 3: There is an inconsistency in these two manuscripts. Bielewicz et al show that mutated 5' splicing site affects miRNA accumulation, while Schwab find that mutated 5' splicing site does not. If possible, the authors shall provide explanation for this inconsistency, which may provides mechanistic insight on how 3' intron affects miRNA accumulation.

Answer 3: Currently, we are working very hard on uncovering the mechanism of the crosstalk between the spliceosome and the miRNA biogenesis machinery. We have already some preliminary data showing that CBP80 (the bigger subunit of the cap-binding protein complex, CBC) interacts with SERRATE which also interacts directly with some of U1 snRNP proteins. Thus, it seems likely that the interaction between CBC, SERRATE and U1 snRNP is crucial for the spliceosome/microprocessor communication. We think that the difference in our and Schwab and colleagues' observations is simply due to the promoter used in both studies: they used the very strong CaMV 35S promoter, and we applied the native *MIR163* promoter. We added this explanation to the Conclusion paragraph of the revised version of our manuscript.

2nd Editorial Decision

15 March 2013

Thank you for the submission of your revised manuscript to our journal. We have now received the enclosed reports from the referees. Referee 2 still has a few suggestions that I would like you to incorporate before we can proceed with the official acceptance of your manuscript.

I think that the splice mutant data could stay in figure 3, however, please perform statistical analyses and tone down the conclusions regarding these mutants. Also, regarding all figures (including supplementary), please specify the error bars, the number of experiments performed (n), and the tests used to calculate p-values in the figure legends. If less than 3 experiments were performed, error bars cannot be shown. In this case, please show the actual data points for each experiment.

In all the figures the miR136 and U6 bands are heavily cropped, which is not acceptable practice. Can you please rerun these gels? The bands should all come from the same gel. In this case, I also would like to strongly encourage you to provide the source data, meaning the entire gels the samples were run on with weight markers, etc. EMBO now offers authors the possibility to publish source data for all figures. Please submit the entire, annotated gels along with the revised manuscript.

For both your paper and the one by Voinnet, the referees point out that the explanation for the discrepancy of the data of both studies is not convincing. Can you, may be, discuss this with Olivier and come to a coherent solution?

I look forward to seeing a new revised version of your manuscript as soon as possible. Let me know please if you have any questions or comments.

REFEREE REPORTS:

Referee #1:

The revised paper by Bielewicz et al. describes data to show and support the interesting phenomenon that introns of plant pri-miRNAs are necessary for biogenesis and function of miRNAs. The authors have done a number of supplementary experiments and, overall, have very well supported their hypotheses. In this version, the paper is clearly written, and the results are arranged in an order that is neat and understandable. I am satisfied that they have addressed my concerns about the experiments.

I did find the quality of the figures to be somewhat lacking, stylistically-speaking, with a mix of text in highly variable sizes that make it difficult or almost impossible to read, some imprecise alignments of text or lines, etc. I would suggest having these remade by a professional.

Referee #2:

In the revised version of the manuscript the authors address several of the comments of the reviewers.

A small statement covering the explanation given in #2 answer 8 concerning the previous detection of At1g66690 in microarray experiments profiling the pathogen response should be included in the manuscript.

Fig. 3:

A line previously designated as an over-expression line of SR33 protein actually has reduced levels of the protein and thus is removed from the revised version. Furthermore, the differences in the splice variants observed in the various SR protein mutants is very small compared to the effect of the cbc mutants, for example. Thus, although they may be statistically significant it is doubtful whether they are biologically relevant. I suggest removing the part of the SR mutants from Fig. 3, especially as the U6 loading control also appears to be lower in these lines compared to the wt reference.

Finally, obviously the authors of this manuscript and the authors of the accompanying manuscript see different reasons to explain the discrepancies between the data reported in the two manuscripts?

Referee #3:

The revised manuscript addressed my concerns.

2nd Revision - authors' response

09 April 2013

Please find enclosed the new revised version of our manuscript entitled "*Introns of plant pri-miRNAs are required for proper biogenesis and function of miRNAs*" by Bielewicz *et al.* This new version not only addresses positively all comments and recommendations of the referees, but also includes all the implementations you have been suggesting in your informal decision letter. Accordingly, we took the following measures:

1. First of all, to improve the quality of the data we have rerun gels and repeated miR163/U6 Northern blots shown in Fig. 1 and 2. We also now provide non-fragmented versions of Northern blots shown in Fig. 3.

2. We have performed statistical analyses of our data, and the results of statistical tests are now included in all figures. These statistical tests have shown that the data presented in the manuscript are statistically significant.

3. As suggested by referee 2, a short statement covering the explanation concerning the previous detection of At1g66690 in microarray experiments profiling pathogen response has now been included in the manuscript..

4. The possible explanations of the discrepancy between our manuscript and that of Schwab et al. were discussed with Rebecca Schwab, and common explanations have been included in both manuscripts.

We hope that both papers, ours and Schwab and colleagues', will be of great interest to the broad readership of *EMBO reports*. Our manuscripts are highly significant and add a novel piece to the current knowledge on regulation of microRNA biogenesis and function.

We confirm that this manuscript has not been published elsewhere and is not under consideration by another journal. All authors have approved the manuscript and agree with its submission to EMBO reports. Thanks for your time and consideration,

3rd Editorial Decision

12 April 2013

I am very pleased to accept your manuscript for publication in the next available issue of EMBO reports. Thank you for your contribution to our journal.

As part of the EMBO publication's Transparent Editorial Process, EMBO reports publishes online a Review Process File to accompany accepted manuscripts. As you are aware, this File will be published in conjunction with your paper and will include the referee reports, your point-by-point response and all pertinent correspondence relating to the manuscript.

If you do NOT want this File to be published, please inform the editorial office within 2 days, if you have not done so already, otherwise the File will be published by default [contact: emboreports@embo.org]. If you do opt out, the Review Process File link will point to the following statement: "No Review Process File is available with this article, as the authors have chosen not to make the review process public in this case."

Finally, we provide a short summary of published papers on our website to emphasize the major findings in the paper and their implications/applications for the non-specialist reader. To help us prepare this short, non-specialist text, we would be grateful if you could provide a simple 1-2 sentence summary of your article in reply to this email.

Thank you again for your contribution to EMBO reports and congratulations on a successful publication. Please consider us again in the future for your most exciting work.